# Fertility in Cystinosis

**DOI:** 10.3390/cells10123539

**Published:** 2021-12-15

**Authors:** Ahmed Reda, Koenraad Veys, Martine Besouw

**Affiliations:** 1Lab of Developmental Biology and Reproductive Medicine, Department of Physiology and Pharmacology, Karolinska Institutet, 17165 Stockholm, Sweden; 2Division of Pediatric Nephrology, Department of Pediatrics, University Hospitals Leuven, 3000 Leuven, Belgium; 3Department of Pediatric Nephrology, University of Groningen, University Medical Center Groningen, 9700 RB Groningen, The Netherlands

**Keywords:** cystinosis, fertility, azoospermia, hypogonadism, cysteamine, histopathology, mouse model

## Abstract

Cystinosis is a rare inheritable lysosomal storage disorder characterized by cystine accumulation throughout the body, chronic kidney disease necessitating renal replacement therapy mostly during adolescence, and multiple extra-renal complications. The majority of male cystinosis patients are infertile due to azoospermia, in contrast to female patients who are fertile. Over recent decades, the fertility status of male patients has evolved from a primary hypogonadism in the era before the systematic treatment with cysteamine to azoospermia in the majority of cysteamine-treated infantile cystinosis patients. In this review, we provide a state-of-the-art overview on the available clinical, histopathological, animal, and in vitro data. We summarize current insights on both cystinosis males and females, and their clinical implications including the potential effect of cysteamine on fertility. In addition, we identify the remaining challenges and areas for future research.

## 1. Introduction

Cystinosis is a rare autosomal metabolic disorder caused by bi-allelic mutations in the *CTNS* gene. This gene encodes the protein cystinosin, which is a lysosomal membrane protein responsible for transporting cystine, produced by the degradation of proteins in lysosomes, from the lysosome into the cytosol. Hence, mutations in *CTNS* cause the intralysosomal accumulation of cystine, leading to various effects in the body [1]. The age at presentation and the severity of the disease allows the classification of cystinosis into three clinical phenotypes: the most severe infantile form (95% of patients), the juvenile form (5%), and the adult ocular benign form (very rare) [2,3]. In the most severe form, the disease initially affects the kidneys, mostly causing end-stage kidney disease (ESKD) in adolescence or early adulthood. In addition, extra-renal complications develop, which most commonly affect the eyes and endocrine and neuromuscular systems [3]. Cysteamine is currently the only available disease-modifying treatment. It is an aminothiol that depletes the accumulated cystine in lysosomes [2,3]. One of the more recently reported unexpected extra-renal complications in male cystinosis patients is azoospermia. Female cystinosis patients, however, have normal fertility and can become pregnant, as reported earlier [4].

For clinical fertility management purposes, azoospermia is classified into obstructive (OA) and nonobstructive azoospermia (NOA), based on clinical sexual characteristics (testicular volume) and sex hormone levels [5]. In OA, the azoospermia is caused by an obstruction in the genital tract and testicular function is preserved, while in NOA, the azoospermia is caused by testicular dysfunction [5]. Hence, azoospermia combined with normal testicular function would most likely be diagnosed as OA, whilst azoospermia combined with impaired testicular function would most likely be diagnosed as NOA [6].

Due to the advances in the treatment of cystinosis patients over recent decades, their life expectancy has substantially increased, making fertility a new and important issue for both patients and their treating physicians [7]. In this review, we provide an overview of the fertility status in cystinosis patients (males and females) and of the possible effects of cysteamine on fertility.

## 2. Fertility in Female Cystinosis Patients

Women with cystinosis have been reported to suffer from delayed puberty, with a menarche around the age of 15–19 years, while a stable menstrual cycle is reached at least two years after menarche [8]. In accordance, the plasma levels of follicle-stimulating hormone (FSH), luteinizing hormone (LH), and estradiol were only rising long after the onset of puberty. However, most adult women with cystinosis showed normal levels of FSH, LH, and estradiol [8].

Unlike men suffering from cystinosis, fertility seems to be unaffected in female cystinosis patients. Several reports have described successful pregnancies in women with cystinosis [4,9,10,11,12,13]. The specific issues that can arise when women with cystinosis become pregnant were recently reviewed by Blakey et al. In summary, various maternal and fetal adverse events were reported, including cephalopelvic disproportion due to maternal short stature, gestational diabetes, hypothyroidism, and respiratory muscle weakness, on top of the complications of a pregnancy following a kidney transplantation [4,14]. The latter include a higher incidence of pre-eclampsia, gestational diabetes, Cesarean section, and pre-term delivery [14].

Another important issue in female cystinosis patients who want children is the teratogenicity of cysteamine. Since this is a lifesaving treatment, the timing of its discontinuation should be well thought out. The current advice is to stop cysteamine as soon as a pregnancy test is found to be positive [13]. However, in one case study, a woman with cystinosis on cysteamine was informed about her pregnancy, being 12 weeks pregnant, but cysteamine treatment had to be discontinued. Later, the patient had a Cesarean section, giving birth to a healthy baby [15]. Since it remains uncertain whether or not cysteamine is excreted into breast milk, breastfeeding is currently not recommended [4].

In general, it is advised for women with cystinosis who want to become pregnant to seek medical advice in advance, preferably in a clinic with expertise in pregnancies in patients with chronic kidney disease or kidney transplantation. By doing so, teratogenic medications can be switched before conception and a plan can be made when cysteamine has to be stopped. Since cystinosis is an autosomal recessive disorder, the chances for the child to develop cystinosis are very low if the father is unaffected. However, since he could be an asymptomatic carrier, pre-conceptional genetic counselling should be offered [16,17]. If the patient’s partner does not carry a mutation in the *CTNS* gene, the chances that the child will develop cystinosis are minimal. Importantly, some mutations can be difficult to detect (such as deep intronic mutations), so couples should be counseled that there is still a very small likelihood that the child will have cystinosis if such a mutation remains undetected in the patient’s partner.

In our expert opinion, we do not advise performing prenatal genetic testing to exclude cystinosis in an unborn child given the risk of miscarriage caused by the procedure. If early genetic testing is warranted, DNA can be extracted from cord blood or from the baby after birth to perform urgent analysis of the *CTNS* gene. The delay of a few weeks in the diagnosis of cystinosis is unlikely to influence long-term prognosis, since the diagnosis will still be made very early, and treatment can be started accordingly.

## 3. Fertility in Male Cystinosis Patients

### 3.1. Sexual Hormone Levels

Primary hypogonadism in male cystinosis patients was first described in 1993 [18]. In those days, treatment with cysteamine was not prescribed and monitored as strictly as it is nowadays, and it was often stopped after kidney transplantation. Hypogonadism with delayed puberty and delayed bone age was found to be very common, but it was not reported whether the men studied (all of whom had been transplanted) were treated with cysteamine or not. Over the years, primary hypogonadism with increased levels of the gonadotropins LH and FSH was reported very frequently in male cystinosis patients [18,19,20,21], and also in those investigated in the current era when treatment with cysteamine is started early in life and continued after kidney transplantation [19,20,21]. In addition, inhibin B is a hormone secreted by Sertoli cells that has an important role in spermatogenesis [22,23]. Since inhibin B was suggested to be a good marker for Sertoli cell function [24], it was added to the panel of hormones to be investigated in more recent studies. Inhibin B levels were found to be reduced in several men [19], a finding that was confirmed in subsequent studies [20,21]. This was a valuable addition to the panel of hormones since for the first time it showed clear evidence for Sertoli cell dysfunction in male cystinosis patients, in addition to Leydig cell dysfunction, which is characterized by low testosterone levels [25].

Reduced kidney function or immunosuppressive treatment for kidney transplantation did not seem to be the cause for these hormone disturbances [18,19,20]. In fact, in the original study by Chik et al. hormone levels in cystinosis patients were compared to a group of men of similar age who underwent a kidney transplant for a disease other than cystinosis, with a comparable renal function and immunosuppressive treatment regimen [18]. Interestingly, secondary hypogonadism, characterized by reduced FSH and LH levels with normal testicular function [18,19], and even normal sex hormone levels have been reported sporadically as well, the latter being more frequently observed in younger men [18,19,20,21].

### 3.2. Testicular Volume and Histology

In the early cohort described by Chik et al., who were likely to be treated with cysteamine less vigorously compared to current practice, testicular volume was reduced in all 10 studied men [18]. In more recent cohorts, it was strikingly found that testicular volume was generally normal in younger men but tended to decrease with increasing age, indicating progressive testicular atrophy [20,21].

There are sporadic reports of testicular biopsy samples, the first being a postmortem investigation showing fibrosis, germinal dysplasia and Leydig cell hyperplasia with numerous cystine crystals, but seminiferous tubules were still visible [18]. Later, histology reports in patients treated with cysteamine confirmed fibrosis, but showed no germinal dysplasia and the Johnson score (a measure for spermatogenesis in the seminiferous tubules [26]) ranged between 7 and 9, indicating intact spermatogenesis [19,20]. Interestingly, while biopsy samples from the central part of the testes showed intact spermatogenesis, more damage was observed, including spermatogonial arrest on light microscopy and enlarged lysosomes in both Leydig and Sertoli cells on transmission electron microscopy in samples taken from the periphery, indicating that the progressive testicular damage seen in male cystinosis patients starts in the periphery at the end of the arterial blood supply [21].

In addition, the infiltration of activated macrophages was found in the interstitial testicular tissues, as well as the presence of perturbed blood–testis barrier in infantile cystinosis patients using histological analysis and Zonula occludens-1 as a marker for the quality of the blood–testis barrier. This could indicate that inflammation might be a common cause for both the primary hypogonadism and for epididymal dysfunction, ultimately causing obstruction [20].

### 3.3. Semen Analysis

In the first paper that mentions semen analysis in cystinosis patients, azoospermia was found in all three investigated men, including one patient with normal sex hormone levels [18].

Years later, semen analysis in five male cystinosis patients confirmed azoospermia in all subjects, even in those with normal sex hormone levels, and normal ejaculate volumes and pH in four of them. In this report, for the first time, concerns were raised regarding male fertility in cystinosis patients treated with cysteamine [19].

Later studies confirmed azoospermia in most men suffering from infantile cystinosis; however, oligozoospermia was found in 1 out of 10 male infantile cystinosis patients in a subsequent investigation [20] and another study in 15 men showed oligozoospermia in two of them, while one man was even reported to have normozoospermia [21]. The ages of these four men with either oligo- or normozoospermia, ranged between 18 and 28 years [20,21]. It remains intriguing as to why this small proportion of men retained viable sperm in their ejaculate, while azoospermia was found in all other men with the infantile phenotype who were of the same age or even younger. The type of *CTNS* mutation did not seem to play a role, since all four men harbored severe mutations in *CTNS* [20,21]. Even more fascinating is the fact that sperm cells could also be demonstrated in the semen of three cystinosis patients with a noninfantile phenotype: oligozoospermia was found in two subjects with juvenile cystinosis (29 and 35 years old), and one man with ocular cystinosis (48 years old) was reported to have normozoospermia and has children [20].

Since male cystinosis patients with a normal Johnson score in their testicular biopsies were often reported to also suffer from primary hypogonadism, this primary hypogonadism could not have been the cause of the observed azoospermia. On the other hand, no sperm could be retrieved by percutaneous epididymal sperm aspiration (PESA) on several occasions in a man with a normal Johnson score, which could indicate a nonobstructive due to his azoospermia. Since testicular ultrasound also showed no signs of obstruction in two additional men, a nonobstructive cause of the azoospermia was initially suspected [19]. However, a few years later, a successful PESA was performed in another male cystinosis patient who had suffered from azoospermia in the previous study [19,27]. The PESA was followed by intracytoplasmic sperm injection (ICSI) and led to the first successful pregnancy induced by a male infantile cystinosis patient, resulting in the birth of healthy twins [27]. Later, viable sperm cells could be extracted by PESA in another patient [20] and by microsurgical testicular sperm extraction (mTESE) in another two men with infantile cystinosis [21], all of whom had azoospermia in their semen analysis, again confirming adequate spermatogenesis in these men. Thus, the evidence for an obstructive cause of cystinosis-related azoospermia with sufficient spermatogenesis started to accumulate.

### 3.4. Scrotal Ultrasound Imaging

Since the finding that there seemed to be no signs of obstruction on testicular ultrasound in two men with cystinosis [19], it has been published how more detailed scrotal ultrasounds could be used to predict obstruction in the genital tract [28]. In order to further investigate the hypothesis of OA in cystinosis, detailed scrotal ultrasound studies were subsequently performed. In one study, it was found that all six male infantile cystinosis patients showed signs of vasal obstruction with an enlarged caput epididymis relative to the ipsilateral testicular volume, of whom one patient with the youngest age had an oligozoospermia (see Figure 1). Interestingly, two male juvenile cystinosis patients with oligozoospermia and one male ocular cystinosis patient with normozoospermia all had a normal scrotal ultrasound [20]. Additionally, in another study, signs of obstruction with dilatation of the rete testis were found in 12 out of 18 investigated male infantile cystinosis patients, two of whom showed oligozoospermia [21].

### 3.5. Seminal Plasma Markers

Several markers in seminal plasma have been studied in order to confirm an obstructive due to the azoospermia observed in infantile cystinosis. One study aimed to perform semen analysis in 15 male infantile cystinosis patients, although semen volume was too small (<0.3 mL) in two of them for seminal markers to be determined. In the remaining patients, normal levels of neutral α-glucosidase (NAG), secreted mainly by epididymis, were found in all except one patient, and the authors concluded that the epididymal secretory capacity and flux of secretions was not affected. They found reduced levels of both fructose (which is the most important source of energy for the spermatozoa and helps to maintain the alkaline pH of the semen), and zinc in 33% of the studied patients. The authors hypothesized that reduced fructose levels could be a sign of obstruction at the level of the excretory ducts of the vesicular glands and that reduced zinc levels could represent an obstruction at the level of the prostate. These abnormalities, however, did not explain all cases of azoospermia or oligozoospermia, since the remaining 67% of patients had normal levels of these seminal plasma markers [21].

In another study, the seminal plasma levels of the epididymal secreted markers Extracellular Matrix protein-1 (ECM-1), which is highly expressed in the epididymis, and NAG, which is a specific and established marker for epididymal secretion used in the previous study as a seminal marker [21], were analyzed. ECM-1 and NAG have been identified as good markers of obstruction of the male genital tract [29,30,31]. These markers were subsequently tested in nine male cystinosis patients (6 with the infantile subtype, 2 with the juvenile subtype and 1 with the ocular subtype) and compared to nine healthy men following vasectomy, and to another seven healthy men without vasectomy. It was found that all infantile cystinosis patients had reduced levels of ECM-1 when compared to healthy controls without vasectomy. Moreover, the levels of NAG were comparable to those in men following vasectomy, which was lower when compared to the levels measured in the men without vasectomy. This seems to contrast to the fact that normal levels of α-glucosidase have been found in the other study of seminal plasma markers of male infantile cystinosis patients. However, the fact that the reported levels were not compared to a normal standard but to two control groups containing healthy men both with and without vasectomy strengthen these results. The combination of reduced ECM-1 and NAG in all studied patients further confirmed the hypothesis of OA as the cause of fertility problems in male infantile cystinosis patients [20] (see Figure 2).

## 4. Fertility in Cystinosis Animal Models

Unequivocally, animal studies can be of great importance to understand mechanisms and pathophysiology of diseases. In this context, male fertility in cystinosis was studied using a *Ctns*^−/−^ knockout mouse model [32] that was generated on a C57BL/6 background, replacing the last four exons of *Ctns* gene with an IRES-β*galneo* cassette [33]. The results showed that male *Ctns*^−/−^ knockout mice showed normal fertility compared to their wild-type litter mates, represented by a similar litter size, testicular morphology, and semen analysis. This meant that the knockout model, albeit showing a renal but mild phenotype, cannot be used to investigate male fertility in cystinosis in absence of the reproductive phenotype [32]. It was not an exception for a mouse model not to exhibit the full clinical phenotype of a disease, since earlier mouse models for similar metabolic disorders, such as Pompe’s disease and Gitelman’s syndrome, could not replicate the full phenotype as well [34,35,36]. Later, it was shown that the male *Ctns*^−/−^ knockout mice had a perturbed blood–testis barrier [20]. A possible explanation behind the absence of the full human phenotype is that the presence of the genetic modifiers can compensate for the absence of a specific gene in mice, and that the renal phenotype in the *Ctns^−/−^* knockout mice, as well as the reproductive phenotype, is less pronounced compared to humans.

In an attempt to find a more suitable animal model for cystinosis, zebrafish were proposed as an alternative [37,38]. In a study by Berlingerio et al., the authors found that both male and female *Ctns*^−/−^ zebrafish were fertile. However, female *Ctns*^−/−^ zebrafish showed reduced egg production and percentage of fertilized eggs compared to wild-type zebrafish [38]. In addition, a histological analysis of the testicular tissue showed the accumulation of spermatozoa in the spermatogenic cysts when compared to wild-type zebrafish [38]. Interestingly, the accumulation of spermatozoa in the spermatogenic cysts could be a sign of obstruction, which potentiate the hypothesis that azoospermia in male infantile cystinosis patients is obstructive in origin.

### In Vitro Studies in Cystinosis 

In addition to these studies on men and mice, a human epididymal epithelial cell line [39] was studied in which the *CTNS* gene was downregulated by siRNA silencing [20]. A transcriptomic analysis revealed altered processes such as fluid shear stress, interleukin-6 production, actin cytoskeleton reorganization, and modified amino acids and sulfur compounds transport [20]. Since these biological processes are important to maintain a healthy epithelial layer, it would be expected that alterations in these processes would lead to a loss of the epithelial layer and shedding of cells, which could play a role in the development of obstruction [20].

## 5. Effect of Cysteamine on Fertility

Cysteamine is a cystine-depleting agent and the only available lifesaving treatment for nephropathic cystinosis. In early studies, it was proven to be a protective agent for spermatogonial stem cells (SSCs) in rats, following X-ray irradiation [40,41]. In addition, studies using cysteamine in sperm cryopreservation in bulls, ram, and goat showed increased sperm motility after thawing [42,43,44]. Nonetheless, there are several reports that suggest a potential negative impact of cysteamine on fertility. When added to the fresh sperm of bulls and buffalos, cysteamine was shown to reduce sperm quality, analyzed by computer assisted sperm analysis system [45,46]. In addition, cysteamine was shown to act as a contraceptive when added to rabbit fresh sperm, inhibiting in vivo fertilization [47]. In one recent study, oral cysteamine treatment in sheep for 6 months at a dose of 20 mg/kg/day reduced sperm count and motility and disturbed blood–testis barrier [48].

In contrast, there are no clear studies on the effects of cysteamine on sperm quality in men. It had been hypothesized that cysteamine could, in theory, cause increased ghrelin levels, which in turn could have a negative effect on Leydig and Sertoli cell function [19]. However, this has never been confirmed. In addition, as explained earlier, over time, the hypothesis of the pathogenesis of azoospermia moved from likely to be nonobstructive towards likely to be obstructive. In addition, when analyzing the age at onset of, and adherence to cysteamine treatment, it was concluded that cysteamine could slow down the testicular degeneration, although it still cannot be fully stopped [21].

Recently, the effect of cysteamine in the previously described cystinosis mouse model was studied, using wild-type C57BL/6 mice and *Ctns*^−/−^ knockout mice that were fed with food mixed with cysteamine at a high dose of 500 mg/kg/day for 6 months [20]. Following the different fertility parameters, no negative effect of cysteamine on epididymal sperm count, litter size, plasma LH, FSH, and testosterone, and seminal vesicle weight was detected in both wild-type and knockout mice. Moreover, a histological analysis of murine testicular tissues showed no effect of cysteamine in both wild-type and knockout mice. The disturbance in the blood–testis barrier that was found in the knockout mice could not be restored by cysteamine. A separate bioavailability study was performed to investigate the availability of cysteamine in testicular tissue, along with cystine accumulation, using wild-type and knockout mice. The results showed that cysteamine did cross the blood–testis barrier but could not reduce cystine accumulation in testicular tissue to the extent that is seen in other organs, such as the kidneys. This could explain to a large extent the absence of an effect of cysteamine on the testis of male cystinosis patients and their fertility in general, even if they were strictly compliant on cysteamine treatment.

In accordance, in a previous study investigating the effect of oral cysteamine on female reproduction in wild-type rats, the authors showed that cysteamine had no adverse effects on conception and early embryonic development [49]. Hence, it is believed that cysteamine would not have a negative impact on reproduction in both male and female cystinosis patients. However, clinical studies investigating the effect of cysteamine in cystinosis patients treated with the drug would have great value.

## 6. Future Perspectives

Even though female infantile cystinosis patients seem to have normal fertility, there is room for future research to focus on neonatal outcomes in these children. To our knowledge, there are no reports of intrauterine growth retardation caused by placental dysfunction in cystinotic women, although in theory placental cystine accumulation could adversely influence its function. However, this has never been studied in a systematic manner. Additionally, it would be interesting to study whether or not compliance to cysteamine treatment prior to pregnancy improves pregnancy outcomes.

Given the obstructive origin of azoospermia, preserving gonadal function in male cystinosis patients, if possible, is of the utmost importance to safeguard the possibility to father their own offspring via assisted reproductive technology. Therefore, future research should be aimed at further elucidating the exact pathogenesis of hypogonadism, which is now presumed to be related to inflammation and fibrosis caused by cystine deposition. More specifically, studying Leydig and Sertoli cell function in cystinosis and how to preserve it, could be of interest. In addition, it remains unclear why regular cysteamine dosages have subtherapeutic effects in testicular tissue and future studies could focus on different treatment strategies that have more of an effect on testicular cystine accumulation. In this regard, a cystinosis animal model more resembling human disease and in vitro models are indispensable.

Additionally, it would be interesting to investigate the inflammatory aspect of cystinosis and its role in the reproductive phenotype, since this could open the door to study the use of anti-inflammatory agents to alleviate the reproductive symptoms. In addition, the use of medications that can lower the seminal plasma viscosity has not been studied thus far.

Furthermore, semen analysis and sperm cryopreservation should be offered to all post-pubertal cystinosis males and hence be included as a recommendation in the standard clinical management guidelines for cystinosis. Once implemented, a large-scale registry-based observational study could provide the most reliable data on the proportion of infantile male cystinosis patients that show oligo- instead of azoospermia and give a better insight into their clinical determinants.

## 7. Conclusions

In conclusion, female cystinosis patients seem to have a normal fertility, although complications during pregnancy that are secondary to the disease and kidney transplantation frequently occur. In contrast, most male infantile cystinosis patients suffer from azoospermia due to obstructive causes, with intact spermatogenesis in early adulthood. This renders them able to father their own biological children using epididymal or testicular sperm, followed by ICSI. Given the fact that cystinosis is an autosomal recessive disorder, the chances for the child to develop cystinosis are extremely small when the partner is confirmed not to be carrier of a *CTNS* mutation.

Surprisingly, oligozoospermia or even normozoospermia can be found sporadically in male infantile cystinosis patients (mainly in those <30 years of age), which may be due to the presence of another yet unidentified factor that mitigates the effect of cystinosis on fertility. Moreover, oligozoospermia and normozoospermia have been reported in juvenile and ocular cystinosis, respectively. Based on this specific finding, we recommend performing a semen analysis at the earliest age possible for (post-pubertal) male cystinosis patients to confirm the presence or absence of azoospermia, since some patients could still have a few sperm cells in their semen which could be cryopreserved. This will increase their opportunities to father their own biological children. Importantly, compliance to cysteamine treatment does not negatively affect the clinical fertility phenotype in male cystinosis patients, albeit it is not able to restore fertility.

## Figures and Tables

**Figure 1 cells-10-03539-f001:**
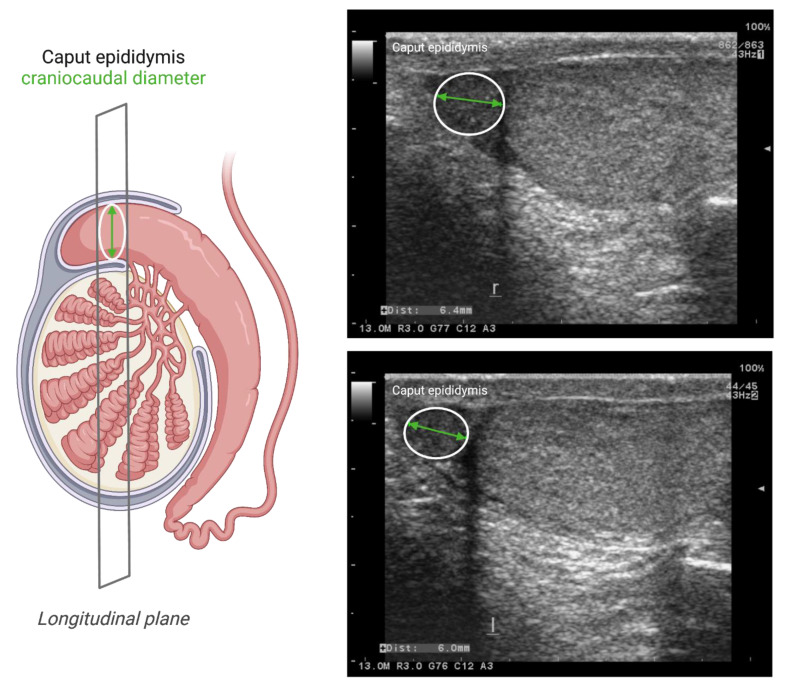
Use of scrotal ultrasound to diagnose obstructive azoospermia. On the left side, a schematic diagram illustrating the hypothesized dilatation in caput epididymis due to obstruction in the male genital tract. The craniocaudal diameter of caput epididymis is determined and normalized to the ipsilateral testicular volume (calculated by the formula radius 1 × radius 2 × radius 3 × 4/3 × π, where radii 1, 2, and 3 are the different axial radii of testis determined by ultrasonic imaging). On the right side, the upper figure shows an example for a scrotal ultrasound image for an adult infantile cystinosis patient, illustrating a dilated caput epididymis. The lower figure shows an example for a scrotal ultrasound image for an adult healthy male, with no signs of dilated caput epididymis [20]. Green arrows in the two ultrasound images represent the craniocaudal diameter of caput epididymis (6.4 mm in upper panel figure; 6.0 mm in lower panel figure).

**Figure 2 cells-10-03539-f002:**
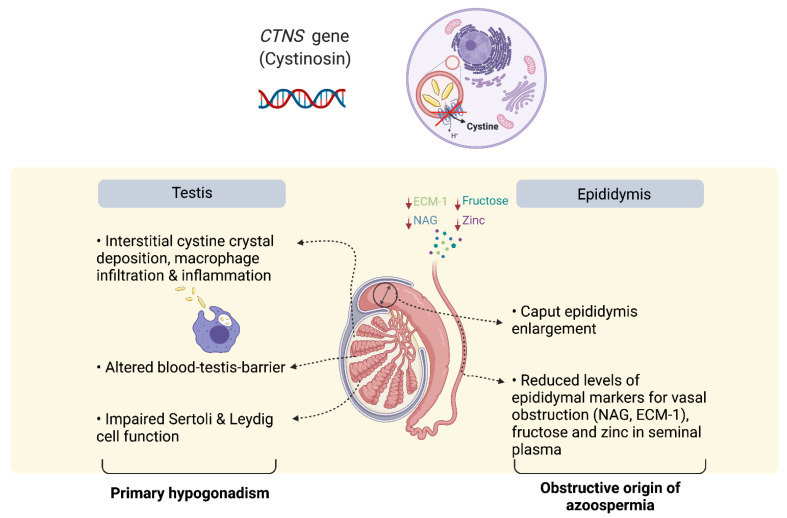
Plausible mechanism of infertility in male infantile cystinosis patients. Bi-allelic mutations in the *CTNS* gene lead to accumulation of cystine in the lysosomes, cell apoptosis, phagocytosis of cell debris and cystine crystals, and inflammation. In kidneys, it is known to be associated with an increased shedding of epithelial cells. In infantile type male cystinosis cystinosis patients, an enlarged caput epididymis, and reduction in seminal plasma markers, including zinc, fructose, extracellular matrix protein-1 (ECM-1) and neutral α glucosidase (NAG), point towards obstruction of the male genital tract as the primary cause for azoospermia. In parallel, cystine accumulation and macrophage infiltration results in interstitial inflammation in the testis, which is associated with alterations at the blood–testis barrier, and Sertoli and Leydig cell impairment, leading to a progressive primary hypogonadism.

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
