# Peer review of "Fertility in Cystinosis"

_cells, 2021, doi:10.3390/cells10123539_

Round 1
Reviewer 1 Report
The authors of the current review manuscript present a very informative summary of infertility status in nephropathic cystinosis. The manuscript is very well designed and written. I have only few suggestions to improve:
- Regarding the fertility in female cystinosis patients, although the reported cases are relatively few, I believe more details need to be added to this section in a trial to answer the following questions. Are there any known sequelae on the newborn from being conceived from a cystinotic mother concerning his early neonatal period and growth parameters afterwards? Is there any data available about difference in the pregnancy outcome if the mother is well controlled compared to poor control in the pre-pregnancy phase? Are there any recommendations for females actively seeking pregnancy but are not pregnant yet? Is there any potential change in their management strategy?
- The concept of prenatal diagnosis following conception in a pregnant cystinotic female should be discussed, whether recommended or not and under which circumstances.
- Figure 1: Can you please add in the legend the dimensions measured by ultrasonography for the caput epididymis in each of the right side images.
- Page 7, line 260: The sentence: "Cysteamine is a cystine-depleting agent and the only available lifesaving treatment nephropathic cystinosis" Please correct the typo and add "….for nephropathic cystinosis".
- You stated that inflammation of both testis and epididymis due to cystine crystal deposition is involved in the pathology stated in both and based on the obstructive azospermic theory, can you please discuss in the future perspective section if you think there is a role for anti-inflammatory drugs or drugs that can lower the seminal viscosity in preserving epididymal integrity in young cystinosis males.
Author Response
Dear reviewer,
Many thanks for reviewing the manuscript and your valuable comments.
- This is indeed an important point. The problems that can arise when women with cystinosis become pregnant were recently reviewed by Blakey et al., this was added in more detail on page 2. However, many of the questions the reviewer poses remain unanswered, since much remains unknown. Hence, we added this in the manuscript on page 8, to give an opportunity for research to cover the unmet demand.
- We have addressed the comment of the reviewer on page 2 and corrected accordingly. A discussion about prenatal diagnosis was added.
- We have addressed the comment of the reviewer on page 5 and corrected accordingly.
- We have addressed the comment of the reviewer on page 7 and corrected accordingly.
- We have addressed the comment of the reviewer on page 8 and corrected accordingly.
Reviewer 2 Report
I thank the authors for this comprehensive review, for which I don't have any criticism.
Just I would talk about infantile-type cystinosis patients.
On page 7: the renal phenotype in the Ctns-/- knockout mice is less pronounced, as well as the fertility phenotype, compared to humans
Author Response
Dear reviewer,
Many thanks for reviewing the manuscript and your valuable comments. We have addressed the comment of the reviewer on page 7 and corrected it accordingly.
Reviewer 3 Report
In my opinion, this Review is focused and well-organized around the subject: fertility in cystinosis.
I have few comments/suggestions for the Authors:
- Please, briefly introduce inhibin B function to facilitate the understanding of the text for non-experts in the field (line 87).
- I found the following sentence too long and not easy to be understood at the first reading. Line 148: “Given no sperm could be retrieved by percutaneous epididymal sperm aspiration (PESA) on several occasions in the man with a Johnson score of 8-9 on testicular biopsy and since testicular ultrasound performed in 2 men showed no signs of obstruction, initially a nonobstructive cause of the azoospermia was suspected”. Please, rephrase it.
- Line 252, the authors state that CTNS gene was downregulated in a human epididymal epithelial cell line in Dube et al (ref: 34). I think this is not correct. Please, check this statement and explain the concept better.
- Line 262: “In addition, studies used cysteamine in sperm cryopreservation in bulls, ram, and goat showed increased sperm motility after thawing.” Would it be “using” instead of “used”. Please, check this sentence.
- Please, remove the sentence line 337-338, since supplementary material is not provided.
Author Response
Dear reviewer,
Many thanks for reviewing the manuscript and your valuable comments.
- We have addressed the comment of the reviewer on page 3 and corrected accordingly. Inhibin function and references were added.
- We have addressed the comment of the reviewer on page 4 and corrected accordingly.
- We have addressed the comment of the reviewer on page 7 and corrected accordingly. In specific, the cell line used was characterized in Dube et al., but CTNS downregulation was performed in Reda et al. hence, we clarified the sentence.
- We have addressed the comment of the reviewer on page 7 and corrected accordingly.
- We have addressed the comment of the reviewer on page 9 and corrected accordingly.